# Characterization of Potential Protein Biomarkers for Major Depressive Disorder Using Matrix-Assisted Laser Desorption Ionization/Time-of-Flight Mass Spectrometry

**DOI:** 10.3390/molecules26154457

**Published:** 2021-07-23

**Authors:** Chieh-Hsin Lin, Hung Su, Chung-Chieh Hung, Hsien-Yuan Lane, Jentaie Shiea

**Affiliations:** 1Department of Psychiatry, Kaohsiung Chang Gung Memorial Hospital, College of Medicine, Chang Gung University, Kaohsiung 833401, Taiwan; cyndi36@gmail.com; 2Graduate Institute of Biomedical Sciences, China Medical University, Taichung 404332, Taiwan; 3School of Medicine, Chang Gung University, Taoyuan 333323, Taiwan; 4Department of Chemistry, National Sun Yat-Sen University, Kaohsiung 804351, Taiwan; impossible122@yahoo.com.tw; 5Department of Psychiatry & Brain Disease Research Center, China Medical University and Hospital, Taichung 404332, Taiwan; alouette_rouge@yahoo.com.tw; 6Department of Psychology, College of Medical and Health Sciences, Asia University, Taichung 413305, Taiwan; 7Department of Medicinal and Applied Chemistry, Kaohsiung Medical University, Kaohsiung 807378, Taiwan

**Keywords:** matrix-assisted laser desorption ionization time-of-flight mass spectrometry, major depressive disorder, schizophrenia, principal component analysis, hierarchical clustering analysis, apolipoprotein C1

## Abstract

Matrix-assisted laser desorption ionization/time-of-flight (MALDI-TOF) mass spectrometry is a sensitive analytical tool for characterizing various biomolecules in biofluids. In this study, MALDI-TOF was used to characterize potential plasma biomarkers for distinguishing patients with major depressive disorder (MDD) from patients with schizophrenia and healthy controls. To avoid interference from albumin—the predominant protein in plasma—the plasma samples were pretreated using acid hydrolysis. The results obtained by MALDI-TOF were also validated by electrospray ionization-quadrupole time-of-flight (ESI-QTOF) mass spectrometry. The analytical results were further treated with principal component analysis (PCA), hierarchical clustering analysis (HCA), and receiver operating characteristic (ROC) curve analysis. The statistical analyses showed that MDD patients could be distinguished from schizophrenia patients and healthy controls by the lack of apolipoprotein C1 (Apo C1), which, in fact, was detected in healthy controls and schizophrenia patients. This protein is suggested to be a potential plasma biomarker for distinguishing MDD patients from healthy controls and schizophrenia patients. Since sample preparation for MALDI-TOF is very simple, high-throughput plasma apolipoprotein analysis for clinical purposes is feasible.

## 1. Introduction

Major depressive disorder (MDD) is a common mental disease that can promote disturbed behavior leading to disability and suicide. Patients may harm themselves, and their condition can affect also others around them and have severe socioeconomic consequences. As the underlying pathophysiology of MDD remains unclear, no objective test is yet available for assisting in its diagnosis or monitoring disease progression. However, some plasma biomolecules have been reported as potential biomarkers for MDD. For example, apolipoprotein D and ceruloplasmin have been reported to be associated with changes in lipid metabolism and immunoregulation in MDD patients [1,2,3]. The concentrations of lipids such as carnitine C10:1 and PE-O 36:5 were shown to be downregulated, while the concentration of LPE 18:1 sn-2 appeared upregulated in MDD patients [4]. Moreover, cholesterol levels were reported to be positively correlated with the severity of depressive symptoms [5]. The concentrations of some amino acids such as tryptophan, lysine, and γ-aminobutyric acid (GABA), were found to be different in MDD patients compared to healthy controls [6]. The glutamic acid level in MDD patients appeared altered after patients underwent selective serotonin reuptake inhibitor (SSRI) treatment [7]. In addition, nervonic acid has also been suggested as a potential biomarker for MDD [8].

Two-dimensional gel electrophoresis (2-DE) and high-performance liquid chromatography/tandem mass spectrometry (HPLC/MS/MS) have been used to perform plasma proteomics and metabolomics studies in the abovementioned research [9,10,11,12,13]. Compared to 2-DE, HPLC/MS/MS has higher reproducibility and sensitivity for characterizing proteins in biological fluids. Though it has been used to detect thousands of proteins in plasma, these traditional analytical approaches (especially 2-DE) require relatively large sample quantities and/or labor- and time-consuming sample preparation that limit their throughput and applicability. It is therefore necessary in the clinic to develop a simple analytical approach that can characterize biological compounds in plasma to distinguish patients with MDD from healthy controls and patients with other mental diseases.

Matrix-assisted laser desorption-ionization time-of-flight (MALDI-TOF) mass spectrometry is a sensitive technique that can characterize biological compounds such as proteins, peptides, and lipids present in sub-femtomole amounts [14,15,16]. The technique is based on soft ionization, hence, MALDI mass spectra are characterized by a very low degree of fragmentation and mostly correspond to singly charged analyte ions. MALDI-TOF is also renowned for its ease of operation [17,18], economical matrix-based sample preparation, and capability for automation, which can facilitate large-scale sample screening. Compared to LC/MS/MS, MALDI-TOF can tolerate much higher salt concentrations, so that biological samples can be rapidly analyzed without undergoing tedious desalting processes. However, some degree of sample preparation is still needed, since the direct MALDI-TOF analysis of plasma without any sample pretreatment will suffer from severe ion suppression effect, whereby the signals of the predominant proteins in the samples will suppress the signals of proteins present at trace level.

To improve the analytical efficiency and avoid the ion suppression effect, a technique involving sample hydrolysis in strong hydrochloric acid followed by MALDI-TOF analysis was developed to decompose the predominant proteins (primarily albumin) in the serum, so that other proteins could be detected [19]. In another study, trifluoroacetic acid was used to effectively hydrolyze albumin, and a decrease was observed in the ion ratio of transferrin to fibrinogen fragments in the sera of patients diagnosed with acute-phase mental disorders [20]. Another analytical strategy involving strong acid hydrolysis followed by off-line HPLC coupled with MALDI-TOF analysis was able to characterize serum IgG levels in schizophrenia patients before and after risperidone treatment [21].

In this study, proteins in plasma samples were precipitated with an organic solvent, after which strong acid hydrolysis followed by MALDI-TOF analysis was used to remove abundant proteins and detect potential biomarkers to distinguish MDD patients from healthy controls and patients with other mental diseases. Since the throughput capacity is the main concern for clinical applications, minimum sample pretreatment was designed and performed, which included diluting plasma with sterilized water, mixing the diluted plasma with the MALDI matrix solution, and air-drying the sample mixture under ambient conditions. The results obtained by MALDI-TOF were validated by electrospray ionization-quadrupole time-of-flight (ESI-QTOF) mass spectrometry to demonstrate the usability of this technique. The analytical results were further analyzed using principal component analysis (PCA), hierarchical clustering analysis (HCA), and receiver operating characteristic (ROC) curve analysis to distinguish the protein profiles of MDD patients, healthy controls, and patients with other mental diseases such as schizophrenia.

## 2. Results and Discussion

### 2.1. Direct Analysis of Plasma Samples by MALDI-TOF Mass Spectrometry

Figure 1 shows the representative MALDI mass spectra from the direct analysis of plasma samples collected from an MDD patient (Figure 1a), a schizophrenia patient (Figure 1b), and a healthy control (Figure 1c). The recorded ions in the presented MALDI mass spectra showed similar patterns, where the only observable ion signals were from albumin ions with +1 to +6 charges. The ion suppression effect is a common problem when using mass spectrometry to analyze mixtures containing molecules characterized by large dynamic ranges of quantity and/or molecular weight, such as serum and plasma. The aforementioned results obviously indicate that direct analysis of plasma using MALDI-TOF would not be suitable to distinguish samples collected from MDD patients, schizophrenia patients, and healthy controls.

### 2.2. Optimization for Acid Hydrolysis of Plasma Samples Followed by MALDI-TOF Analysis

The interference of albumin in plasma during MALDI-TOF analysis may be avoided by either: (1) removing albumin through laborious and time-consuming liquid chromatographic, affinity, or electrophoretic separation or (2) selectively decomposing albumin using chemical or biochemical reagents, which may be a time- and labor-saving approach when using a strong acid. Previously, acid hydrolysis was proved to be a useful method to rapidly decompose serum album; however, an optimal protocol for acid hydrolysis to remove plasma albumin must be established to obtain reproducible MALDI-TOF results. Trifluoroacetic acid solutions prepared in three different dilutions (50%, 25%, and 10%) were used to examine the efficiency of acid hydrolysis for the detection of the precipitated plasma proteins. The MALDI mass spectrum (inset of Figure 2a–c) showed that albumin in the plasma was efficiently removed by TFA hydrolysis. This confirmed our previous studies. Figure 2a–c show the MALDI mass spectra of plasma samples collected from a healthy control and hydrolyzed with TFA for 10 min. The results showed that acid hydrolysis using a 25% TFA solution through the V_sample_/V_matrix_ ratio at 1:1 gave the highest signal-to-noise ratio (S/N) in the *m/z* 2000–8000 range (Figure 2b). Compared to other different V_sample_/V_matrix_ ratios (1:2, 1:5, 1:10, 1:20, 1:50, and 1:100), the crystallization can be homogenously formed at the ratio of 1:1. Since sweet spot will be seriously affected the quality of mass spectra recorded from MALDI analysis, it will be helpful for further automated MALDI analysis. In addition, the MALDI target plate will be damaged (acid corrosion) when using higher concentration (50%) of TFA solution.

The selection of an appropriate matrix is crucial for obtaining high-quality MALDI mass spectra. Three common MALDI matrices—α-CHC, 2,5-DHB, and SA—were used to examine the protein ion signals via MALDI-TOF analysis. One microliter of acid-hydrolyzed plasma solution was mixed with a different volume of the MALDI matrix solution and then deposited on the stainless-steel target plate. After air-drying, the sample spot was irradiated using a pulsed Nd-YAG laser beam, and the desorbed ions were detected in a TOF mass analyzer. Three different sample/analyte volume ratios (1:1, 1:5, and 1:10, *v*/*v*) were tested. As shown in Figure 3, the α-CHC matrix produced the best-quality mass spectra with regard to ion sensitivity within a mass range of *m*/*z* 2000–8000 (Figure 3a). The parameters for protein precipitation (ACN), acid hydrolysis (25% TFA solution), and matrix preparation (α-CHC, V_sample_:V_matrix_ = 1:1) for MALDI-TOF analysis were then optimized and used for all plasma sample analyses throughout the entire study.

### 2.3. Optimized MALDI-TOF Analysis for Plasma Samples

Figure 4 displays the representative MALDI mass spectra (*m/z* 2000–8000) of the plasma samples collected from three MDD patients (Figure 4a–c), three schizophrenia patients (Figure 4d–f), and three healthy controls (Figure 4g–i). Obviously, the ion profiles of the plasma samples from MDD patients were quite different from those of the schizophrenia patients and healthy controls. As it can be seen, the ion signals detected for the samples from healthy controls and schizophrenia patients show a much higher relative intensity than those for the samples of patients with MDD in the mass range of *m/z* 6000–7000. In addition, the ions around *m/z* 2600, 2800, and 4900 show different intensity trends for MDD patients, schizophrenia patients, and healthy controls, allowing an easy distinction of the three groups.

### 2.4. Principal Component Analysis

Principal component analysis factors and compares all analyte ion signals detected among the samples. The PCA score plot reflects the differences of the ion patterns of the samples [22], while the PCA loading plot provides information on the contribution of each ion signal to the variance covered by its respective principal component. Figure 5 displays the PCA results for the plasma samples obtained from MDD patients, healthy controls, and schizophrenia patients. Two separated groups were obtained in the PCA score plots when analyzing the data of the samples collected from 20 MDD patients (red) and 20 healthy controls (green) (Figure 5a). Figure 5b displays the PCA loading plot shown in Figure 5a. The three ions (*m/z* 6630, 3314, and 6432) are characteristic of the samples collected from healthy controls, while the two ions (*m/z* 2742 and 2862) are characteristic of the samples collected from patients with MDD. These abovementioned ions contribute mainly to group separation in the PCA plot. No significant difference was found between the data of the samples collected from the 20 schizophrenia patients (blue) and those of the samples from the 20 healthy controls (green) (Figure 5c). Figure 5d displays the PCA loading plot shown in Figure 5c. Only the two ions (*m/z* 4961 and 2742) are partially characteristic of the samples collected from schizophrenia patients. Figure 5e displays the PCA results for the 60 total samples collected in this study. It is evident that the ion profiles obtained for the MDD patients are well-separated from those obtained for the healthy controls but are indistinguishable from some of the profiles recorded for the schizophrenia patients. Figure 5f displays the PCA loading plot shown in Figure 5e. The ions at *m/z* 6630, 3314, and 6432 are the main contributors to the group separation in the PCA plot. Obviously, the ion at *m/z* 3314 is the doubly charged ion (M^2+^) at *m/z* 6630. These ions are characteristic for the samples collected from healthy controls and schizophrenia patients; therefore, their signals are useful for distinguishing healthy controls and schizophrenia patients from MDD patients.

### 2.5. Hierarchical Clustering Analysis

Hierarchical clustering analysis, a grouping method that functions by creating a cluster tree or dendrogram, is a popular multivariate analysis method [23] that was used in this study to evaluate the relationship between healthy controls and MDD and schizophrenia patients by looking for similarities in the detected ion signals in their MALDI mass spectra. As shown in Figure 6, MDD patients and healthy controls are clearly separated from each other, confirming the results obtained by PCA. However, the ion cluster for patients with schizophrenia was not clearly separated from that of healthy controls and MDD patients. These results suggest that some schizophrenia and MDD patients may share common biomarkers.

### 2.6. Receiver Operating Characteristic Curve Analysis

The data collected from MALDI-TOF analysis were further analyzed using ROC curve analysis. Three ROC curves for healthy controls and MDD patients were obtained in accordance with the aforementioned multivariate statistical analyses. As shown in Figure 7, the area under the curves (AUC) ranges between 0.80 and 0.86, reflecting the obvious differences in sensitivity and specificity among the samples. These results also indicate that the ions at *m*/*z* 6630 and 6432 are potential biomarkers for distinguishing MDD patients from healthy controls and schizophrenia patients. The results obtained by ROC curve analysis further confirmed those obtained using PCA and HCA.

### 2.7. Biomarker Identification and Clinical Implications

The identification of the peptide or protein ion signals (*m/z* 6630 and 6432) detected on the MALDI-TOF mass spectra is a challenging task, since these molecules either are present in low abundance or have high molecular weights, making it nearly impossible to identify them using MALDI-TOF/TOF. A combination of the standard addition approach and literature search was implemented to assist in the identification of potential biomarkers detected by acid hydrolysis followed by MALDI-TOF analysis. The literature search results indicated that some apolipoproteins with similar mass range as the potential biomarkers observed in this study have been reported as potential biomarkers for psychiatric diseases. The levels of such apolipoproteins (i.e., Apo A1, Apo A2, Apo A4, Apo C1, and Apo D) were significantly decreased in the serum of schizophrenia patients in a study analyzing the proteome of serum samples collected from first-onset drug-naive patients and healthy volunteers by LC/MS/MS [24]. A serum-based test developed by Rules-Based Medicine, using with DiscoveryMAP multiplex immunoassay profiling, showed significant changes in the levels of Apo A2, Apo B, and Apo C1 in bipolar disorder patients compared to healthy controls [25]. Human Apo C1 is a component of lipoproteins and is encoded by the APOC1 gene [26,27]. This gene is expressed primarily in the liver and is activated when monocytes differentiate into macrophages. Apolipoprotein C1 is composed of 57 amino acids (inset in Figure 8b), is abundantly found in plasma, and is responsible for the activation of esterified lecithin cholesterol, with an important role in the exchange of esterified cholesterol between lipoproteins and in the removal of cholesterol from tissues. It is one of the most positively charged proteins in the human body [28]. Its main function is the inhibition of cholesteryl ester transfer protein, which probably occurs through the alteration of the electric charge of high-density lipoprotein (HDL) molecules. Since Apo C1 is associated with glycol metabolism and lipid metabolism, several studies have revealed the role of Apo C1 in other non-psychiatric diseases. For instance, Apo C1 was significantly upregulated in the serum of women with polycystic ovary syndrome with respect to that of healthy controls, as shown by surface-enhanced laser adsorption/ionization (SELDI) bio-chip analysis [29]. The protein and mRNA levels of Apo C1 and Apo E in the brain were also analyzed by enzyme-linked immunosorbent assay (ELISA) and RNase protection assay, indicating their correlation in Alzheimer’s disease patients [30]. Even though acid-hydrolyzed plasma, instead of other biofluids, was analyzed in this study, apolipoproteins may still be useful biomarkers for distinguishing samples collected from patients with different mental diseases.

Based on the abovementioned literature search results, another analytical approach, i.e., standard addition, for specific compound identification was used to compare the protein ion signals detected from an Apo C1 protein standard solution and plasma samples. Figure 8a shows the MALDI mass spectrum of the Apo C1 standard which was purchased from Sigma-Aldrich. Unexpectedly, at least four protein ion signals (M_1_ to M_4_) were detected in the spectrum. To further confirm the presence of multiple proteins in the Apo C1 standard, ESI-QTOF was used to analyze the standard solution. Again, several protein ion signals (M_1_, M_3_, and M_4_) were detected in the deconvoluted ESI mass spectrum (inset in Figure 8a).

The proteins detected in the protein standard may be: (1) full-length (M1) Apo C1 and truncated (M_2_) Apo C1 isoforms that concurrently exist in the standard, so that the mass difference (*m*/*z* = 6630 − 6432 = 198 amu) between the peaks for each respective isoform corresponds to the loss of Thr–Pro—the expected cleavage of the N-terminal dipeptide; there are no other known modifications that are consistent with this mass change (Figure 8b) [31]; (2) impurities or salt adducts formed during synthesis and purification of the protein standard (i.e., M_3_ and M_4_). The MALDI mass spectrum of TFA-hydrolyzed plasma samples collected from MDD patients (red), healthy controls (green), schizophrenia patients (blue), and of two solutions obtained by spiking an MDD patient’s plasma sample with different concentrations of Apo C1 (10^−5^ M, black; 10^−6^ M, gray) are shown in Figure 8b. The results indicate that two of the Apo C1 ion signals (*m/z* 6630 and 6432) matched well with those detected for the plasma collected from healthy controls and schizophrenia patients (M_1_ and M_2_, blue and green) but not with those found for the plasma from MDD patients (red). The results suggest that Apo C1 (especially, the full-length isoform at *m/z* 6630 and 6432) may be a useful biomarker for distinguishing healthy controls and schizophrenia patients from MDD patients.

## 3. Materials and Methods

### 3.1. Plasma Samples Collection and Storage

One hundred and twenty individuals were recruited with the approval of the institutional review board of China Medical University Hospital, Taichung, Taiwan (CMUH104-REC2-081). The study was conducted in accordance with the current revision of the Declaration of Helsinki. Twenty patients with MDD (10 males, 33.2 years old on average; 10 females, 29.1 years old on average) and 20 healthy controls (10 males, 32.9 years old on average; 10 females, 29.1 years old on average) were selected. For comparison, 20 patients (10 males, 33.0 years old on average; 10 females, 29.0 years old on average) with chronic schizophrenia were also recruited. Basic information of subjects was listed in Table 1. Patients and healthy controls were evaluated by psychiatrists according to the Diagnostic and Statistical Manual of Mental Disorders-4th (DSM-IV) criteria using the Structured Clinical Interview for DSM-IV [32,33]. All the participants were between 19 and 48 years of age, physically healthy, and presented all routine laboratory assessments (including blood, biochemical, and electrocardiography tests) within healthy limits. Subjects were excluded if they had a DSM-IV diagnosis of schizoaffective disorder or bipolar disorder, mental retardation, substance abuse or dependence, epilepsy or a history thereof, had head trauma or CNS diseases other than schizophrenia, and were pregnant or lactating.

All patients with MDD had major depressive episodes with a 17-item Hamilton Depression Rating Scale score of 18 or higher. All patients with schizophrenia had been stabilized with antipsychotics for more than three months. The healthy controls did not have any Axis I or II psychiatric disorders. Five milliliters of peripheral venous blood was collected into an EDTA-containing blood collection tube to obtain plasma. The blood tube was centrifuged at 1500× *g* for 10 min at 4 °C. Plasma was obtained immediately and frozen in −80 °C freezer before analysis. Once plasma samples were obtained from the subjects in each group, these samples were immediately delivered to the laboratory and divided into several parts (100 μL/tube) to avoid the influence of freeze-and-thaw on biological samples, and then stored at −80 °C until analysis.

### 3.2. Chemicals and Materials

The MALDI matrices—α-cyano-4-hydroxycinnamic acid (α-CHC), 2,5-dihydroxybenzoic acid (2,5-DHB), and sinapinic acid (SA)—were purchased from Sigma-Aldrich (St. Louis, MO, USA). The solvents methanol (MeOH) and acetonitrile (ACN) were purchased from Merck (Darmstadt, Germany). Acetic acid (AA) and trifluoroacetic acid (TFA) was purchased from Sigma-Aldrich. An apolipoprotein C1 (Apo C1) standard was purchased from Sigma-Aldrich. Matrix solutions were prepared in 70% ACN containing 0.1% TFA (*v*/*v*) at different concentrations for MALDI-TOF analysis (α-CHC: 10 mg/mL; 2,5-DHB and SA: 20 mg/mL). Distilled deionized water (purified with a PURELAB Classic UV from ELGA, Marlow, UK) was used to prepare the electrospray solution for ESI-QTOF analysis. All chemicals and HPLC-grade solvents were used without further purification.

### 3.3. Acid Hydrolysis

In order to standardize the procedures of acid hydrolysis, an aliquot (20 μL) of plasma sample was added into an Eppendorf tube and mixed with 40 μL of ACN to precipitate proteins. The solution was then centrifuged at 10,000× *g* for 5 min at 25 °C. After removing the supernatant, 20 μL of TFA solution (50%, 25%, or 10%) was added to the respective pellet for hydrolysis. After hydrolysis for 10 min and centrifugation (10,000× *g*, 5 min, 25 °C), the supernatant was orderly collected to another new Eppendorf tube to make sure the TFA-hydrolyzed plasma pellet treated the same duration. The hydrolysis-supernatant was then mixed with matrix solution at different ratios (V_sample_:V_matrix_ = 1:1, 1:2, 1:5, 1:10, 1:20, 1:50, 1:100) and analyzed by MALDI-TOF analysis.

### 3.4. Mass Spectrometric Analyses

One microliter of each acid-hydrolyzed plasma sample solution was deposited on a spot on a MALDI target plate and mixed with an optimal volume of a MALDI matrix solution. After air-drying, the sample plate was transferred into a MALDI-TOF (AutoFlex III, Bruker Daltonics, Leipzig, Germany) ionization source operated using the FlexControl 2.2 software (Ver. 3). While in the MALDI source, each sample spot was irradiated with a pulsed Nd-YAG laser (355 nm) for desorption and ionization. Two thousand laser shots were averaged per sample to obtain a representative mass spectrum. Triplicate analysis was determined in each sample. To avoid sweet spot effect occurred in MALDI-TOF analysis and import the high-quality mass spectra into subsequent statistical software, MALDI mass spectra were recorded under manual operation at the beginning of this study. Random walk mode was then applied to collect MALDI data based on the optimized V_sample_/V_matrix_ ratios (1:1) for further analysis. The positive ion MALDI mass spectrum was recorded in linear mode at an acceleration voltage of 20 kV in delayed extraction mode. External calibrations were performed using horse apomyoglobin (*m/z* 16,952), bovine cytochrome c (*m/z* 12,361), insulin (*m/z* 5734), and ACTH 18–39 (*m/z* 2466) standards. The tolerance of mass error was set ±100 ppm in this study. The experimental procedure is shown in Figure 9a–h.

A quadrupole time-of-flight mass spectrometer (micrOTOF-QII, Bruker Daltonics, Bremen, Germany) equipped with an ESI source was used to detect the molecular ions of Apo C1 in standard solution. The operating parameters of the QTOF were as follows: capillary entrance voltage 4.5 kV, nebulizer gas 12 L/min, dry gas flow rate 4 L/min, and dry gas heating temperature 250 °C. The electrospray solution (50% MeOH with 0.1% AA, *v*/*v*) was infused through a fused-silica capillary (i.d. 100 μm) at a flow rate of 120 μL/h. The mass spectrometer was operated in positive ion mode over a scan range of *m/z* 500–1500 and calibrated from *m/z* 118 to *m/z* 2721 using a tuning mix solution from Bruker Daltonics.

### 3.5. Multivariate Statistical Analyses

The MALDI-TOF results of the samples collected from healthy controls, MDD patients, and schizophrenia patients were further analyzed and compared using PCA, HCA, and ROC curve analysis. PCA and HCA were performed using the ClinProTools 2.2 software (ion peak selection) from Bruker Daltonics, while ROC curve analysis was performed using the SPSS 12.0 software package (SPSS, Chicago, IL, USA). The utilities of this software include ion peak definition (signal-to-noise ratio = 3; relative threshold base peak = 1% in total average spectrum), baseline subtraction (Convex Hull Baseline, 0.8% Baseline Flatness), intensity normalization, peak picking, and statistical testing (Wilcoxon/Kruskal-Wallis and *t* test/analysis of variance). A *p*-value less than 0.05 was defined as statistically significant. Receiver operating characteristic curve analysis was used to examine the specificity and sensitivity of the biomarkers to distinguish between samples from MDD patients, healthy controls, and schizophrenia patients. The correlation between true-positive (sensitivity) and false-positive rates (specificity) is presented. All experimental procedure is shown in Figure 9a–h.

## 4. Conclusions

In this study, we pretreated and analyzed plasma samples collected from patients diagnosed with MDD and healthy controls, as well as from schizophrenia patients. Directly analyzing plasma samples is not an effective strategy because of the high degree of ion suppression from the predominant plasma proteins. An alternative approach involving the use of ACN precipitation, TFA-hydrolysis, MALDI-TOF analysis, and multivariate statistical analyses was used to characterize plasma protein/peptide biomarkers for distinguishing MDD patients from healthy controls and schizophrenia patients. Two potential ion signals (*m/z* 6432 and 6630) were detected and served as tentative biomarkers through standard addition approach for distinguishing MDD patients and healthy controls. A downregulated level of Apo C1 was found in the analyte profiles of plasma samples collected from MDD patients, indicating that Apo C1 is a potential biomarker for distinguishing MDD patients from healthy controls and schizophrenia patients. The developed MALDI-TOF approach to detect the presence of protein/peptide biomarkers in plasma is advantageous as it requires a simple sample treatment and short analysis time and provides objective results. In addition, the high degree of automation of MALDI-TOF analysis allows for high-throughput analysis, so that a large number of clinical samples can be efficiently screened. The development of such a biochemical analytical approach would be helpful for assessing psychiatric disorders. Since Apo C1 is an important biological compound involved in the inflammatory and immune responses, these mechanisms may also be involved in the progression of MDD. Therefore, further identified work and studies are necessary for understanding the role of Apo C1 in MDD.

## Figures and Tables

**Figure 1 molecules-26-04457-f001:**
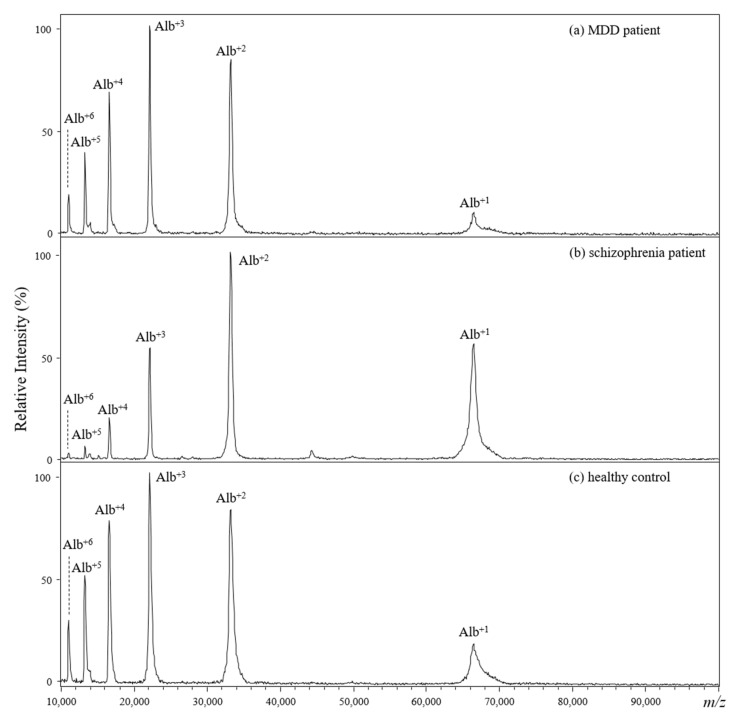
(**a**–**c**) Positive-ion MALDI mass spectra of diluted plasma from (**a**) an MDD patient, (**b**) a schizophrenia patient, and (**c**) a healthy control. Alpha-CHC served as the MALDI matrix, which was mixed with each diluted plasma solution.

**Figure 2 molecules-26-04457-f002:**
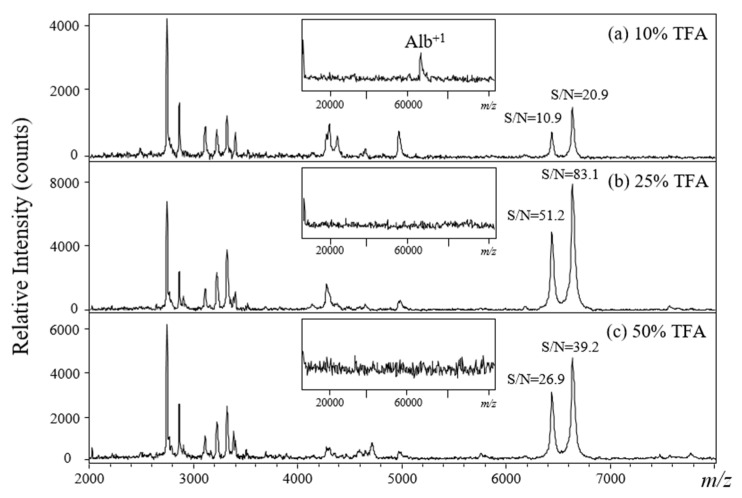
(**a**–**c**) Positive-ion MALDI mass spectra for plasma samples hydrolyzed with different TFA acid solutions. MALDI matrix: α-CHC in 50% ACN incorporating 0.5% TFA (V_sample_:V_matrix_ = 1:1).

**Figure 3 molecules-26-04457-f003:**
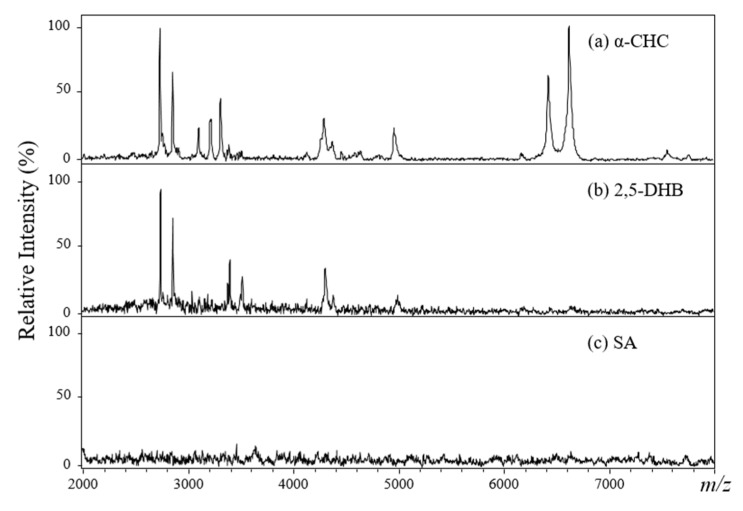
(**a**–**c**) Positive-ion MALDI mass spectra for 25% TFA-hydrolyzed plasma samples from a healthy control using different matrices (V_sample_:V_matrix_ = 1:1).

**Figure 4 molecules-26-04457-f004:**
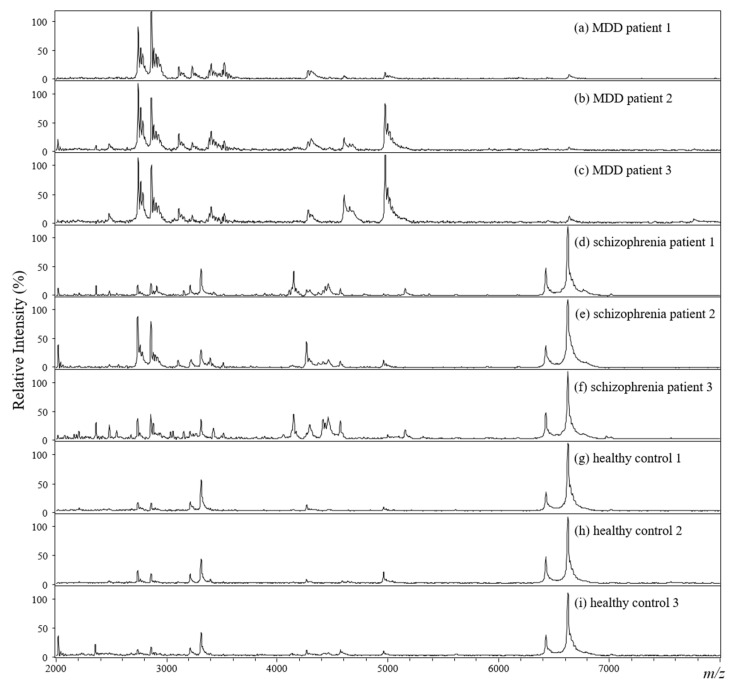
MALDI mass spectra of plasma samples from (**a**–**c**) MDD patients, (**d**–**f**) schizophrenia patients, and (**g**–**i**) healthy controls after hydrolysis with 25% TFA for 10 min and mixture with the α-CHC matrix (V_sample_:V_matrix_ = 1:1).

**Figure 5 molecules-26-04457-f005:**
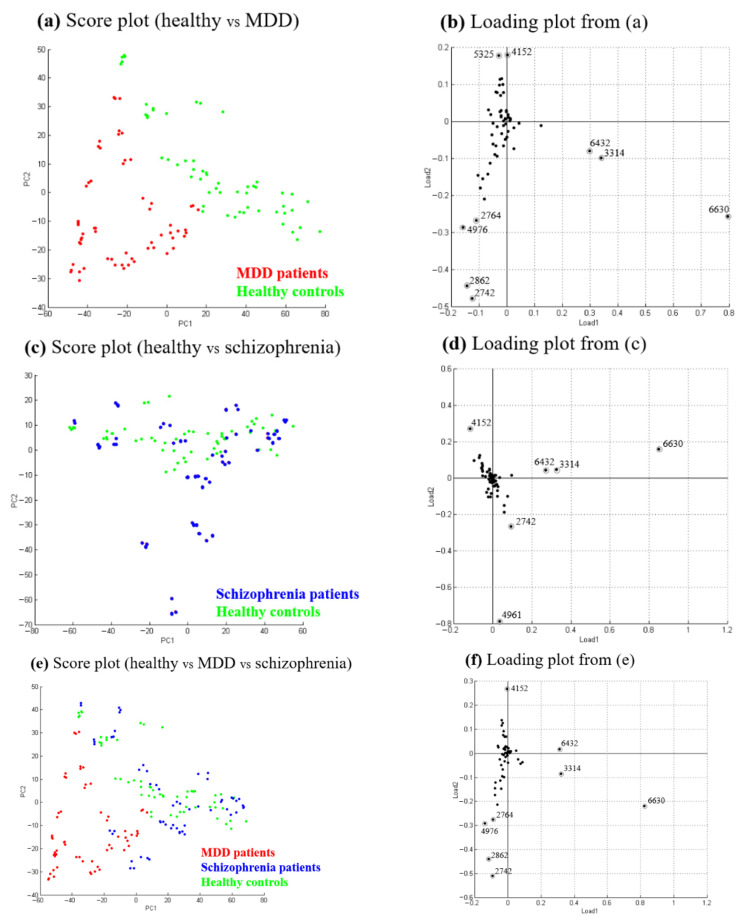
PCA score plot (**a**–**c**) and PCA loading plot (**d**–**f**) for MALDI-TOF data from acid hydrolysis-treated plasma samples derived from healthy controls, MDD patients, and schizophrenia patients aged between 19 and 48 years: (**a**,**b**) healthy controls and MDD patients, (**c**,**d**) healthy controls and schizophrenia patients, and (**e**,**f**) healthy controls, MDD patients, and schizophrenia patients.

**Figure 6 molecules-26-04457-f006:**
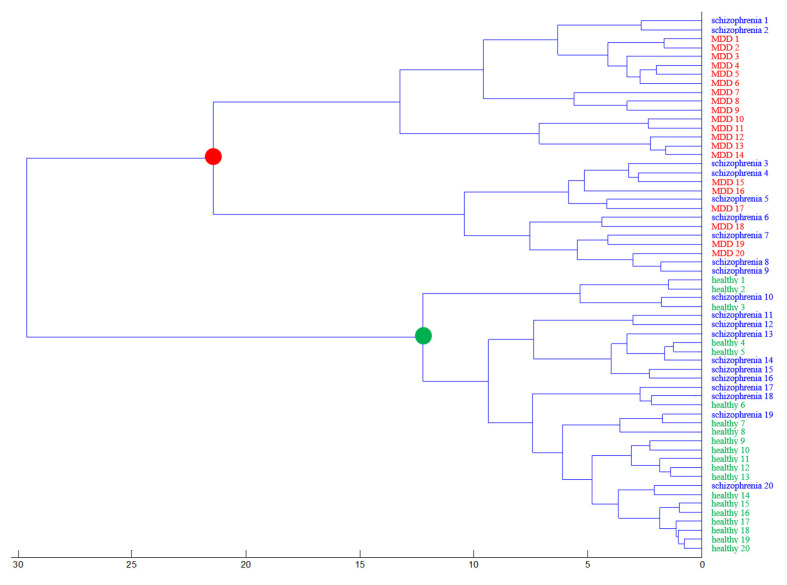
Results of hierarchical clustering analysis based on PCA results.

**Figure 7 molecules-26-04457-f007:**
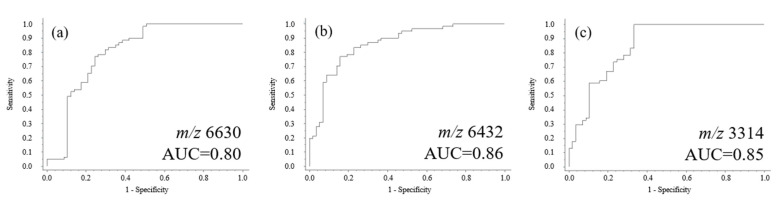
Representative ROC curves (**a**–**c**) analysis for healthy controls and MDD patients. The AUC of possible protein/peptide ions ranges between 0.80 and 0.86.

**Figure 8 molecules-26-04457-f008:**
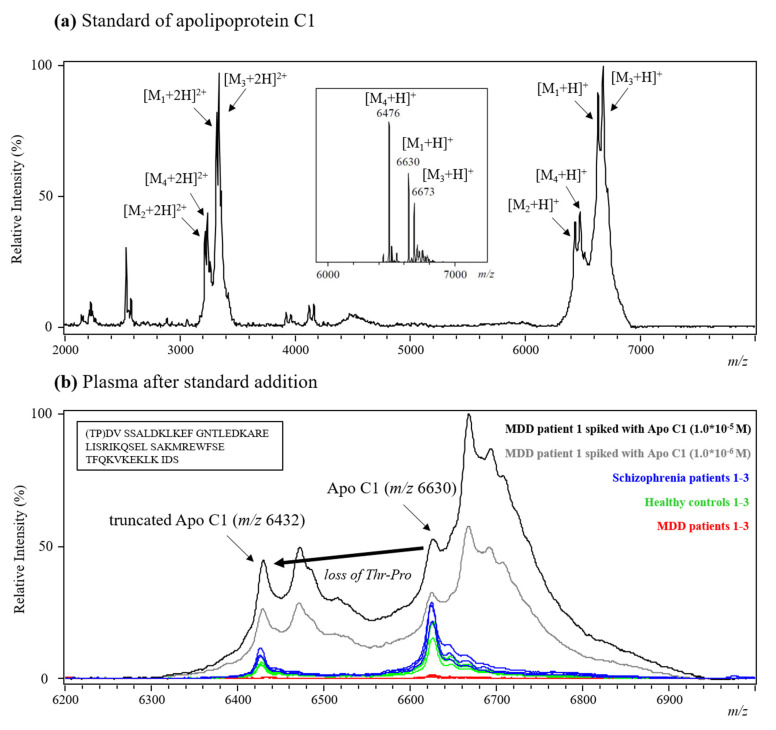
(**a**) The positive MALDI spectrum of the Apo C1 standard (inset: deconvoluted ESI-QTOF mass spectrum of Apo C1 standard). (**b**) MALDI mass spectra recorded from the analysis of TFA-hydrolyzed plasma collected from three schizophrenia patients, three healthy controls, three MDD patients, and one MDD patient’s spiked with the Apo C1 standard (inset: amino acids sequence of Apo C1).

**Figure 9 molecules-26-04457-f009:**
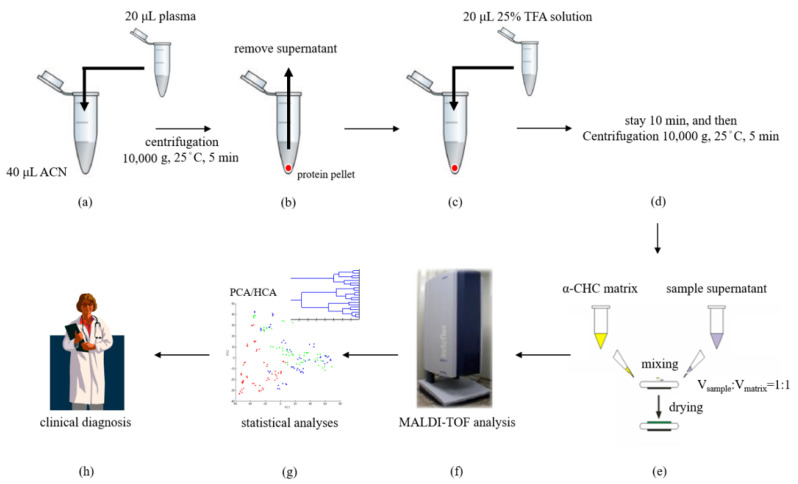
Overview of the MALDI-TOF analytical approaches: (**a**) an aliquot (20 μL) of plasma sample was added into an Eppendorf tube that contained ACN (40 μL) and (**b**) was immediately vortexed and centrifuged at 10,000× *g* for 5 min at 25 °C for protein precipitation. (**c**) The pellet was acid-hydrolyzed via treatment with 25% TFA (20 μL). (**d**) After 10 min acid-hydrolysis and centrifugation (10,000× *g*, 5 min, 25 °C), the supernatant was collected and then (**e**) mixed with the MALDI matrix to form crystals. (**f**) The sample crystals were irradiated with a pulsed Nd-YAG laser for desorption and ionization. Acquisition of protein/peptide profiles via MALDI-TOF analysis was followed by (**g**) characterization of the plasma samples using multivariate statistical analyses to facilitate (**h**) clinical diagnosis.

**Table 1 molecules-26-04457-t001:** Clinical data of the subjects used in this study.

Healthy			Schizophrenia			Major Depressive Disorder		
Code	Gender	Age	Code	Gender	Age	Code	Gender	Age
HCBC6-1	Male	23	BE34-1	Male	29	CNN48-1	Male	42
HCBC9-1		24	BE38-1		42	CNN49-1		26
HCBC39-1		40	BE41-1		34	CNC56-1		20
HCBC53-1		37	BE81-1		37	CNC74-1		47
HCBC93-1		40	BE125-1		21	CNC77-1		37
HCBC106-1		21	BE202-1		47	CNC78-1		32
HCBC129-1		35	BE207-1		39	CNC90-1		22
HCBC139-1		48	DRM104		23	CNC95-1		42
HCBC152-1		29	DRM109		33	CNC106-1		29
HCBC153-1		32	RNMak-1		25	CNC107-1		35
*n* = 10		32.9 (avg.)	*n* = 10		33.0 (avg.)	*n* = 10		33.2 (avg.)
HCBC3	Female	22	BE66-1	Female	29	CNN38-1	Female	26
HCBC15-1		26	BE71-1		32	CNN43-1		25
HCBC20-1		19	BE201-2		26	CNN50-2		28
HCBC31-1		27	BED008-1		19	CNC52-1		41
HCBC41-1		34	BED009-1		26	CNC60-1		37
HCBC110		36	DRM102		21	CNC63-1		32
HCBC131-1		28	DRM103		27	CNC73-1		23
HCBC135-1		26	DRM110		36	CNC79-1		26
HCBC141-1		32	RNMg		34	CNC88-2		34
HCBC151-1		41	RNNt-1		40	CNC89-1		19
*n* = 10		29.1 (avg.)	*n* = 10		29.0 (avg.)	*n* = 10		29.1 (avg.)
20 (total)		31.0 (F + M)	20 (total)		31.0 (F + M)	20 (total)		31.15 (F + M)

## Data Availability

The data presented in this study are available in this article.

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
