# Peer review of "Characterization of Potential Protein Biomarkers for Major Depressive Disorder Using Matrix-Assisted Laser Desorption Ionization/Time-of-Flight Mass Spectrometry"

_molecules, 2021, doi:10.3390/molecules26154457_

Round 1

Reviewer 1 Report

Major issues,

  1. Authors used MALDI-TOF to analyze TFA-hydrolyzed plasma samples and claimed that two signals which belongs to ApoC1 can be used as biomarker to distinguish healthy and schizophrenia patients. However, how comes all other proteins in plasma were hydrolyzed during the TFA treatment but not ApoC1? Authors need to provide a clear and strong evidence to prove that those two signals (6432 and 6630) are ApoC1.
  2. Since ApoC1 is very important in glycan and lipid metabolism, authors also mentioned that some previous studies also indicated that ApoC1 may be the biomarker in other diseases (2.7 biomarker, starting from line 215), then how could this common protein ApoC1 can be the single biomarker for MDD if those two signals (6432/6630) are ApoC1?
  3. The TFA-hydrolysis is the critical step in sample preparation, and it can significantly affect the results. How to standardize the procedures and how to carry out this step of quality control?

Minor issues,

  1. Authors mentioned that 3 different sample/analyte ratio were tested (line 142), however, Fig 3 shows MS result for 1 ratio and author doe not provide sufficient info about this data.
  2. The representative MALDI spectra (Fig 4) are highly similar in MDD or schizophrenia patients. However, PCA in Fig 5 shows quite different spreading results. Author should provide basic info about patients, such as gender, age etc. in this study.
  3. Wrong labeling in Fig 5b.
  4. If 6432 and 6630 signals can be used as biomarkers, is there any threshold value or any other reference signal to distinguish MDD and healthy?

Reviewer 2 Report

Manuscript presented by Lin and co-workers entitled “Characterization of Potential Protein Biomarkers for Major Depressive Disorder Using Matrix-Assisted Laser Desorption Ionization/Time-of-Flight Mass Spectrometry” presents the MALDI-TOF was used to characterize potential plasma biomarkers for distinguishing patients with major depressive disorder (MDD) from patients with schizophrenia and healthy controls. Manuscript is good written however, due to the methodological drawbacks, not adequately described results and conclusions not supported by the data,  before publication in Molecules, paper needs major revision.

My major comments are presented below. Provide the explanation for all of them, make changes in the text.

Major concerns:

- All of the text – once Authors use Fig another one Figure in the text. Please unify.

- Figure 4 caption – remove italic

- Figure 5 a, b,c – the presented plots suggest that the control group was different in each analysis because there are significant differences in the values presented in the Fig 5 a, b and c when it comes to the healthy control group. Please correct or discuss/explain

- Figure 5b – schizophrenia patients data are presented in blue however there are only two spots, additionally other data in red are presented. This diagram is different from a, and c.

- Direct analysis of plasma samples by MALDI-TOF analysis – page 11, line 277 – the title of this section does not correspond to the text presented below. The text treats about the plasma sample collection and storage.

- What was the volume of collected sample? What volume was used for single experiment? How many times each experiment was repeated?

- Describe the plasma sample preparation

- Did you analyze the influence of sample storage (time) on the obtained results?

- The experimental procedure is shown in Figs. 1a-h. – page 12, line 330 – this is not fig 1 – this is fig 9.

- in your procedure did you compare the results obtained for the samples containing TFA and without it? Did you try to evaporate TFA before sample analysis?

- on most  figures Authors use relative intensity – only on Figure 8 there are different units on y axes however Authors still call them as relative intensity. Correct.

- In my opinion good idea will be not to use relative intensity in the case of figure 2. According to this figure, could you explain the influence of TFA added to the sample on the signal intensities presented on the spectrum? There is a lot of reports giving information about the effect of TFA on the MALDI mass spectra. Why did the Authors analyze only solution containing 10, 25 and 50% of this acid?

- in their investigation Authors based only on the application of internal standard in the form of Apoliporpotein C1, as based on the literature data, may corresponds to  the signals observed at m/z 6630 and 6432. Why Authors did not try to determine the chemical structure of this protein present in plasma samples using proteomics methods?

- what was the temperature of centrifugation (Fig 9, text – page12, line 315)?

- correct language, errors, style

- what was the accuracy of MS analyses?

 - Check and correct the reference style (lack of page numbers etc.)

Reviewer 3 Report

Review on
Characterization of Potential Protein Biomarkers for Major Depressive Disorder Using Matrix-Assisted Laser Desorption Ionization/Time-of-Flight Mass Spectrometry

In this article, a MALDI-TOF method has been developed to differentiate patients with major depressive disorder, from patients with schizophrenia and healthy controls. Acid hydrolysis was applied prior to MALDI-TOF analysis, different matrices were tested. The evaluation of the results was carried out by statistical analysis including PCA (principal component analysis) ROC (receiver operating characteristic curve analysis) and HCA (hierarchical clustering analysis) methods. The article is well organized only some additional details should be added (see the comments). The results are promising and they are presented properly. 
I suggest minor revision before acceptance. 

Comment:
1.    What do you mean by “relatively soft” ionization technique, in the introduction? (page 2, line 68)

2.    On page 4 lines 131-132, the hydrolysis of plasma compounds (from healthy peoples) are compared based on their signal-to-noise ratio, however, the numbers are not given. Please add the S/N ratio for some peaks in figure 2, in order to make it easier to compare. 

3.    Page 4 line 142. The authors applied different matrices and different sample/matrix ratios. The results are not discussed. It is not written which ratio resulted in the best spectra. It must be detailed in the article. Furthermore, the MALDI method development is an important part of the article, thus the spectra obtained at different matrices and ratios (matrix/analyte) should be shown in the supporting information, indicating the S/N values- These data are important for the readers who want to develop or apply similar method.

4.    Page 4 lines 143 to 147 The alpha CHC matrix is mentioned as the best matrix for such an application. The next sentence is the following: “The parameters for protein precipitation, acid-hydrolysis, and matrix preparation for MALDI-TOF analysis were then optimized and used for all plasma sample analyses throughout the entire study.” 
However, the optimal parameters are not discussed. It must be detailed which method was optimal based on what criteria.
(The comment corresponds to the previous one (3.).)

5.    In the caption of Figure 3, the applied matrix ratio should be added. 

Statistical analysis
6.    The data selection should be detailed, what peak picking parameters were applied for the extraction of data sets? The Mass spectra in Figure 4 are quite different for all three groups. What is the reason that the statistical analysis cannot find differences between healthy and schizophrenic patients? How the error (mass accuracy) of the external calibration was involved?
In figure 5 on the loading plot some m/z values with high accuracy. Such a high accuracy was achieved?

7.    Page 12 line 330, Figs 1a-h is mentioned, however, such figures are not presented in the article. 

8.    In the conclusion part, the authors highlight the advantage of automation of MALDI-TOF analysis. Were the measurements carried out manually or automatically? It should be mentioned in the experimental part. 

Round 2

Reviewer 1 Report

1. Authors used the standard addition method and then claimed that two signals at m/z 6432 and 6630 are from ApoC1 (Fig. 8). Since MDD patient's plasma have low signal intensity at m/z 6432 and 6630, why not use MDD patient's plasma supplemented with ApoC1, and then performed the complete experiment procedure (TFA-hydrolysis and MALDI-TOF) to prove the existence of m/z 6432 and 6630 are from ApoC1?

2. Unlike most biomarkers distinguish abnormal conditions by “increasing” the signal, authors suggested that the downgraded MS signals at 6432 and 6630 could be the biomarkers to distinguish MDD. In addition, authors also mentioned that they used “manual operation” but not the typical “random walk” mode while collecting MALDI data. Manual operation is very likely to introduce unnecessary human error, causing 6432 and 6630 to be undetected.

Reviewer 2 Report

The revised version of the presented manuscript meets all of my requirements. Authors gave answers and discussed all of my points.

Check spelling and corrects small errors. 

Manuscript can be accepted for publication.

Author Response

Errors have been modified in the revised manuscript. Thank you very much.